# Long Non-Coding RNA *KCNQ1OT1* Regulates Protein Kinase CK2 Via miR-760 in Senescence and Calorie Restriction

**DOI:** 10.3390/ijms23031888

**Published:** 2022-02-08

**Authors:** Yoonsung Lee, Young-Seuk Bae

**Affiliations:** BK21 FOUR KNU Creative BioResearch Group, School of Life Sciences, Kyungpook National University, Daegu 41566, Korea; lyslys8621@naver.com

**Keywords:** *KCNQ1OT1*, long non-coding RNA, senescence, calorie restriction, protein kinase CK2, miR-760

## Abstract

Long non-coding RNAs (lncRNAs) play important biological roles. Here, the roles of the lncRNA *KCNQ1OT1* in cellular senescence and calorie restriction were determined. *KCNQ1OT1* knockdown mediated various senescence markers (increased senescence-associated β-galactosidase staining, the p53-p21^Cip1/WAF1^ pathway, H3K9 trimethylation, and expression of the senescence-associated secretory phenotype) and reactive oxygen species generation via CK2α downregulation in human cancer HCT116 and MCF-7 cells. Additionally, *KCNQ1OT1* was downregulated during replicative senescence, and its silencing induced senescence in human lung fibroblast IMR-90 cells. Additionally, an miR-760 mimic suppressed *KCNQ1OT1*-mediated CK2α upregulation, indicating that *KCNQ1OT1* upregulated CK2α by sponging miR-760. Finally, the *KCNQ1OT1*–miR-760 axis was involved in both lipopolysaccharide-mediated CK2α reduction and calorie restriction (CR)-mediated CK2α induction in these cells. Therefore, for the first time, this study demonstrates that the *KCNQ1OT1*–miR-760–CK2α pathway plays essential roles in senescence and CR, thereby suggesting that *KCNQ1OT1* is a novel therapeutic target for an alternative treatment that mimics the effects of anti-aging and CR.

## 1. Introduction

Long non-coding RNAs (lncRNAs) are transcripts with a size greater than 200 nucleotides that are not translated into proteins. LncRNAs are responsible for diverse functions, including transcriptional regulation in *cis* or *trans*, organization of nuclear domains, and interaction with microRNAs (miRNAs) [1,2,3]. In particular, understanding the interaction network of lncRNAs and miRNAs provides a new perspective on the regulatory mechanisms of genes. The lncRNA *KCNQ1OT1*, the full name of which is *KCNQ1* overlapping transcript 1, is a 91 kb transcript that is transcribed by RNA polymerase II in an antisense orientation, relative to *KCNQ1* [4,5]. *KCNQ1OT1* is present in intron 10 of the KCNQ1 gene. The *KCNQ1* locus is located on the short arm of human chromosome 11 (11p15.5). Through recruiting G9a and the H3K27 histone methyltransferase PRC2 (polycomb repressive complex 2), *KCNQ1OT1* interacts with chromatin to form a complex folding structure and then silences multiple target genes [6]. Additionally, *KCNQ1OT1* is involved in several disorders, including heart disease, cerebral ischemic stroke, atherosclerosis, pyroptosis, and various cancers (gastric, ovarian, and colorectal) by binding to various miRNAs (for example, miR-760, miR-200a, miR-452-3p, miR-320a, miR-701-3p, and miR-2054), all of which regulate the expression of their target genes [7,8,9,10,11,12,13]. Nevertheless, the biological roles of *KCNQ1OT1* in senescence and calorie restriction (CR) remain unclear.

Cellular senescence is the terminal arrest of proliferation, stimulated by several cellular stresses such as telomere shortening, oncogenic activation, and oxidative stress [14,15]. Previous studies identified the protein kinase CK2 (CK2), composed of two catalytic (α and/or α′) subunits and two regulatory β subunits, as a senescence regulator. CK2 inhibition triggers the expression of several senescence markers, including senescence-associated β-galactosidase (SA-β-gal) activity [16], p53–p21^Cip1/WAF1^ axis activation [17], reactive oxygen species (ROS) production [18], senescence-associated heterochromatin foci (SAHF) formation [19], and senescence-associated secretory phenotype (SASP) expression [20]. miR-186, miR-216b, miR-337-3p, and miR-760 promote cellular senescence by inhibiting CK2α [21,22]. Calorie restriction (CR), consisting of a chronic reduction in total calorie intake without malnutrition, is the most successful strategy to delay cellular senescence [23]. It has recently been reported that CK2 is upregulated by CR and induces autophagy [24]. However, the molecular mechanism underlying CR-mediated CK2 upregulation remains unclear.

In this study, the potential role of *KCNQ1OT1* in senescence and CR was assessed. *KCNQ1OT1* knockdown promoted a senescence phenotype via downregulating CK2α in human cancer MCF-7 and HCT116 cells, and lung fibroblast IMR-90 cells. This research indicates that *KCNQ1OT1* upregulates CK2α expression through interaction with miR-760 during senescence and CR, suggesting that *KCNQ1OT1* may be a novel therapeutic target for aging-associated diseases.

## 2. Results

### 2.1. KCNQ1OT1 Knockdown Induced Activation of SA-β-gal Staining, the p53-p21^Cip1/WAF1^ Pathway, and H3K9 Trimethylation Via CK2α Silencing in Human Cancer Cells

MCF-7 and HCT116 cells were transfected with *KCNQ1OT1* siRNA to investigate the involvement of *KCNQ1OT1* in senescence. *KCNQ1OT1* knockdown upregulated SA-β-gal activity (Figure 1A). Additionally, immunoblot results indicated that *KCNQ1OT1* knockdown upregulated the levels of p53 and p21^Cip1/WAF1^, and hallmarks of SAHF (increased H3K9 trimethylation (H3K9me3) and decreased H3K9 acetylation (H3K9Ac)) (Figure 1B). It was previously reported that CK2 downregulation induces these senescence markers (activation of SA-β-gal staining, the p53-p21^Cip1/WAF1^ pathway, and SAHF) [16,17,19]. Therefore, it was examined whether *KCNQ1OT1* interacts with CK2. Interestingly, ectopic CK2α expression abrogated the induction of SA-β-gal activity, p53, p21^Cip1/WAF1^, and H3K9me3, mediated by *KCNQ1OT1* downregulation (Figure 1A,B). Furthermore, *KCNQ1OT1* knockdown reduced the protein level of CK2α (Figure 1B). These results collectively suggest that *KCNQ1OT1* knockdown induces cellular senescence by downregulating CK2. 

### 2.2. KCNQ1OT1 Knockdown Induced SASP Factor Expression and ROS Generation Via CK2α Silencing in Human Cancer Cells

It was investigated whether *KCNQ1OT1* downregulation increased the expression of SASP factors due to reports that senescent cells secrete pro-inflammatory factors [14,15]. *KCNQ1OT1* knockdown induced the expression of SASP factors, including interleukin (IL)-1β, IL-6, and matrix metalloproteinase (MMP) 3 (Figure 2A). *KCNQ1OT1* downregulation increased the amount of intracellular ROS because oxidative stress is a major cause of senescence [14,15]. For this purpose, HCT116 and MCF-7 cells were transfected with *KCNQ1OT1* siRNA and stained with CM-H_2_DCFDA. *KCNQ1OT1* knockdown increased ROS production, as indicated by the right shift in fluorescence during flow cytometry (Figure 2B). It was previously reported that CK2 downregulation induces SASP expression [20] and ROS generation [18]. Ectopic expression of CK2α abrogated the induction of SASP factor expression and ROS generation, mediated by *KCNQ1OT1* downregulation (Figure 2A,B). Collectively, these results indicate that *KCNQ1OT1* knockdown induces ROS generation and inflammation through downregulating CK2.

### 2.3. KCNQ1OT1 Was Involved in Lipopolysaccharide (LPS)-Mediated SASP Factor Expression Via Silencing CK2α in Human Cancer Cells

Because treatment with LPS causes cellular senescence by downregulating CK2α [20], it was tested whether LPS downregulated *KCNQ1OT1* expression. Treatment with LPS (6 μg/μL) reduced the transcript levels of *CK2α* and *KCNQ1OT1* in human cancer cells (Figure 3A). Furthermore, the effect of *KCNQ1OT1* on SASP factor expression in cells treated with LPS was investigated. However, treatment with LPS (6 μg/μL) increased SASP factor (IL-1β, IL-6, and MMP3) expression in human cancer cells; additional treatment with pcDNA3.1-KCNQ1OT1 (36,181–37,140) abrogated the LPS-mediated induction of SASP factors (Figure 3B). It was previously shown that the concerted action of miR-760, miR-186, miR-337-3p, and miR-216b stimulated premature senescence through silencing the CK2α protein in HCT116 cells [21], and that miR-760 and miR-186 were upregulated in replicative senescent IMR-90 cells [22]. The expression patterns of these miRNAs affected by treatment with LPD were determined. Quantitative real-time polymerase chain reaction (qPCR) analysis revealed that the amount of miR-760 increased by more than 200% in both HCT116 and MCF-7 cells treated with LPS (6 μg/μL), in comparison with the control cells. miR-186 was not upregulated by LPS treatment, and miR-337-3p and miR-216b were differently regulated in these cells (Figure 3C). Thus, these results collectively indicate that LPS increased miR-760 amounts via downregulating *KCNQ1OT1*, resulting in CK2α downregulation-mediated senescence.

### 2.4. KCNQ1OT1 Upregulated CK2α by Sponging miR-760 in Human Cancer Cells

Next, it was examined whether *KCNQ1OT1* regulates CK2α expression via miR-760. HCT116 and MCF-7 cells were transfected with pcDNA3.1-KCNQ1OT1 (36,181–37,140) or KCNQ1OT1 siRNA, along with an miR-760 mimic or inhibitor (sequences shown in Appendix A). Ectopic expression of *KCNQ1OT1* (36,181–37,140) increased the mRNA level of *CK2α*, whereas additional treatment with the miR-760 mimic suppressed *KCNQ1OT1*-mediated *CK2α* upregulation (Figure 4A). In contrast, *KCNQ1OT1* knockdown reduced the mRNA level of *CK2α*, whereas additional treatment with an miR-760 inhibitor suppressed *KCNQ1OT1* knockdown-mediated *CK2α* downregulation (Figure 4B). Both the miR-760 mimic and inhibitor could not change the amount of *KCNQ1OT1*, indicating that miR-760 was not an upstream regulator of *KCNQ1OT1*. Altogether, these results indicate that *KCNQ1OT1* increases the amount of *CK2**α* mRNA by sponging miR-760. Appendix A shows the sequences and binding sites of *KCNQ1OT1*, miR-760, and *CK2**α* mRNA, determined using TargetScan and miRanda.

### 2.5. CR Condition Upregulated CK2α by miR-760 Downregulation Via KCNQ1OT1 Upregulation in Human Cancer Cells

It was supposed that *KCNQ1OT1* expression was upregulated in the CR condition because the level of *CK2α* mRNA was upregulated in the CR condition [24]. HCT116 and MCF-7 cells were incubated in CR conditions to test this hypothesis. As shown in Figure 5A, the expression of *KCNQ1OT1* and *CK2α* was induced by CR in these cells, whereas additional treatment with *KCNQ1OT1* siRNA suppressed CR-mediated *CK2α* induction, indicating *KCNQ1OT1* as a positive regulator of *CK2α* in CR conditions. Additionally, analysis with real-time qPCR indicated that the level of miR-760 decreased by 70% in CR conditions, whereas the levels of miR-186, miR-216b, and miR-337-3p in CR conditions were unchanged or increased (Figure 5B). Next, the effect of miR-760 on CR-mediated *CK2α* upregulation was examined. Treatment with an miR-760 mimic suppressed CR-mediated *CK2α* upregulation, indicating miR-760 as a major negative regulator of *CK2α* in CR conditions (Figure 5C). Finally, real-time qPCR analysis revealed that treatment with *KCNQ1OT1* siRNA increased the levels of miR-760, whereas CR suppressed *KCNQ1OT1* knockdown-mediated miR-760 induction (Figure 5D). Altogether, these results indicate that CR downregulated miR-760 by upregulating *KCNQ1OT1*, resulting in *CK2α* upregulation in human cancer cells. Because it has been reported that the lncRNA *SNHG6* also acts as a sponge of miR-760 in CRC cells [25], it was tested whether CR upregulates *SNHG6* expression. However, *SNHG6* expression was unchanged in the CR condition (data not shown).

### 2.6. KCNQ1OT1 Was Downregulated during Replicative Senescence in Human Lung Fibroblast Cells, Which Was Rescued by CR Conditions

Since CK2α is downregulated in replicative senescent cells and aged tissues [16], it was examined whether *KCNQ1OT1* is downregulated during senescence in human lung fibroblast IMR-90 cells. *KCNQ1OT1* knockdown decreased the mRNA level of *CK2α* in IMR-90 cells (Figure 6A). Conversely, *KCNQ1OT1* knockdown upregulated SA-β-gal activity in IMR-90 cells, and ectopic expression of CK2α abrogated the induction of SA-β-gal activity mediated by *KCNQ1OT1* downregulation (Figure 6B). To determine how *KCNQ1OT1* expression decreased by replicative senescence, IMR-90 cells were repeatedly passaged until a senescence-like state was observed. Most cells at PDL 47 stained positively for SA-β-gal, whereas only a few stained positively for SA-β-gal among early-passage (PDL 34) cells (data not shown). The transcript levels of *KCNQ1OT1* decreased by 60% in replicative senescent cells (PDL 47), compared to early-passage (PDL 34) cells, indicating that *KCNQ1OT1* was downregulated during replicative senescence. Finally, CR conditions increased *KCNQ1OT1* by 150% in early-passage (PDL 34) cells compared with normal calorie conditions. However, CR conditions increased *KCNQ1OT1* more strongly (by 250%) in replicative senescent cells (PDL 47) compared with normal calorie conditions, indicating that CR can rescue the decreased expression of *KCNQ1OT1* and *CK2α*, mediated by replicative senescence (Figure 6C). Collectively, these data indicate that replicative senescence decreases *CK2α* expression via downregulating *KCNQ1OT1*, and that CR can suppress replicative senescence via a *KCNQ1OT1*–CK2 axis.

## 3. Discussion

Epigenetic regulation mechanisms, including microRNAs (miRNAs), nucleosome remodeling, DNA methylation, and histone modification, can change heritable phenotypes without changing the DNA sequence [26]. There are two main classes of non-coding RNAs (ncRNAs): the larger long ncRNAs (lncRNAs; >200 nucleotides), and the smaller miRNAs (21–25 nucleotides) [1,2,3]. *KCNQ1OT1* was initially identified as an lncRNA, which silences several genes within the kcnq1 locus via establishing a higher-order structure of chromatin [4,5,6]. Recently, other biological functions of *KCNQ1OT1* have been reported, including those associated with fibrosis, cerebral ischemia, stroke atherosclerosis, oncogenesis, osteogenic differentiation, fracture healing, and cardiac hypertrophy [7,8,9,10,11,12,13]. However, the detailed mechanisms by which *KCNQ1OT1* regulates cellular senescence and CR are still unknown. Therefore, we attempted to clarify whether *KCNQ1OT1* was involved in senescence and CR. This study shows that *KCNQ1OT1* knockdown induced several cellular senescence markers, including induction of SA-β-gal activity, activation of the p53-p21^Cip1/WAF1^ pathway, increased H3K9 trimethylation required for SAHF formation, and SASP factor (IL-1β, IL-6, and MMP3) expression, in human cancer HCT116 and MCF-7 cells (Figure 1 and Figure 2A). Because oxidative stress is a significant cause of senescence [14,15], the effect of *KCNQ1OT1* on ROS generation was investigated. As expected, *KCNQ1OT1* knockdown increased intracellular ROS in HCT116 and MCF-7 cells (Figure 2B). Additionally, the level of *KCNQ1OT1* mRNA was reduced in LPS-induced senescent cells, and ectopic expression of *KCNQ1OT1* suppressed LPS-mediated SASP factor expression in these cancer cells (Figure 3A,B). Furthermore, *KCNQ1OT1* knockdown induced SA-β-gal activity in human lung fibroblast IMR-90 cells, and *KCNQ1OT1* expression was decreased during replicative senescence in IMR-90 cells (Figure 6). Collectively, these results indicate that *KCNQ1OT1* acts as an anti-aging reagent and antioxidant in human cancer cells and lung fibroblasts.

It was previously shown that CK2 knockdown induces senescence markers such as SA-β-gal staining [16], activation of the p53-p21^Cip1/WAF1^ pathway [17], ROS generation [18], SAHF formation [19], and SASP factor expression [20]. Therefore, the relationship between *KCNQ1OT1* and CK2 was examined. This study shows that CK2α overexpression abrogated these senescence markers caused by *KCNQ1OT1* knockdown (Figure 1, Figure 2, and Figure 6B). Additionally, *KCNQ1OT1* knockdown decreased the protein and mRNA levels of CK2α (Figure 1B, Figure 2A, and Figure 4B), and *KCNQ1OT1* (36,181–37,140) overexpression stimulated the mRNA level of *CK2α* (Figure 4A), indicating that *KCNQ1OT1* functions as a positive regulator of *CK2α* in cells. Recent studies have revealed that lncRNAs can sponge miRNAs, inhibiting the expression of target mRNA [1,2,3]. Because previous studies have shown that miR-760 interacts with *CK2α* [21,22] and *KCNQ1OT1* [27], a signaling network of *KCNQ1OT1*, *CK2α*, and miR-760 was examined. This study shows that additional treatment with an miR-760 mimic suppressed *KCNQ1OT1*-mediated *CK2α* upregulation, and additional treatment with an miR-760 inhibitor suppressed *KCNQ1OT1* knockdown-mediated *CK2α* downregulation (Figure 4). Consistently, treatment with LPS, which decreased *KCNQ1OT1* expression, increased the level of miR-760 (Figure 3C). Hence, these studies suggest that the amount of *KCNQ1OT1* RNA can titrate miR-760 and thereby regulate miR-760 availability for binding to the mRNA of *CK2α*. *KCNQ1OT1* and CK2α can crosstalk through their transcript’s ability to compete for miR-760 binding.

Because CR provides an anti-aging effect [23], how CR regulates *KCNQ1OT1* and miR-760 was examined. This study shows that the CR condition increased the expression of *KCNQ1OT1* and *CK2α* and decreased the level of miR-760 (Figure 5A,B). Furthermore, treatment with an miR-760 mimic suppressed CR-mediated *CK2α* upregulation (Figure 5C). Consistently, *KCNQ1OT1* knockdown increased the levels of miR-760, whereas additional CR conditions suppressed *KCNQ1OT1* knockdown-mediated miR-760 induction (Figure 5D). Finally, the CR condition rescued the decreased expression of *KCNQ1OT1* and *CK2α*, mediated by replicative senescence (Figure 6C). Altogether, these results reveal that the CR condition downregulated miR-760 via upregulating KCNQ1OT1, resulting in CK2α upregulation in human cells. Although lncRNA *SNHG6* acts as a sponge of miR-760 in CRC cells [25], CR did not change the *SNHG6* expression under experimental conditions (data not shown). How does CR upregulate *KCNQ1OT1* expression? Other groups have previously demonstrated that the transcription factors NFY and β-catenin regulate *KCNQ1OT1* [28,29]. However, CR could not regulate the expression of NF-Y and β-catenin, at least under experimental conditions (data not shown). Thus, additional studies are required to identify a specific upstream regulator of *KCNQ1OT1* in CR conditions.

In conclusion, on the basis of this study’s results, a model for the role of *KCNQ1OT1* in senescence and CR was proposed. *KCNQ1OT1* upregulates *CK2α* by sponging miR-760. The *KCNQ1OT1*–miR-760–CK2 pathway is involved in senescence and CR. LPS treatment and cell division in fibroblasts induce senescence by downregulating *KCNQ1OT1*. CR suppresses senescence by activating the *KCNQ1OT1*–miR-760–CK2 pathway (Figure 6D). Therefore, this study suggests *KCNQ1OT1* as a novel therapeutic target in aging and age-related diseases, as well as an alternative treatment that mimics the effects of CR.

## 4. Materials and Methods

### 4.1. Materials

Antibodies against CK2α (#sc-373894), p53 (#sc-71820), p21^Cip1/WAF1^ (#sc-6246), and β-actin (#sc-47778) were obtained from Santa Cruz Biotechnology (Santa Cruz, CA, USA). Antibodies specific for histone H3- (#ab1791) and histone H3 Lys9 trimethylation (H3K9me3) (#ab8898) were purchased from Abcam (Cambridge, England). Antibodies specific for H3 Lys9 acetylation (H3K9Ac) (#06-942), 5-bromo-4-chloro-3-indolyl-β-D-galactoside, and LPS were obtained from Sigma Chemical Co. (St. Louis, MO, USA). CM-H_2_DCFDA was obtained from Invitrogen (Carlsbad, CA, USA).

### 4.2. Cell Culture and CR

Human diploid fibroblast IMR-90 cells were obtained from the American Type Culture Collection (ATCC, Manassas, VA, USA) at a population doubling level (PDL) of 24. Human breast cancer MCF-7 cells, colon cancer HCT116 cells (ATCC), and IMR-90 cells were cultured in Dulbecco’s modified Eagle’s medium containing 10% (*v*/*v*) fetal bovine serum and 1% (*v*/*v*) penicillin-streptomycin, in a humidified atmosphere of 95% (*v*/*v*) air and 5% (*v*/*v*) CO_2_ at 37 °C. The PDL of IMR-90 cells was calculated using the formula PD = log(*Nf/Ni*)/log2, where *Nf* is the final cell number, and *Ni* is the initial number of seeded cells. For cell starvation (calorie restriction), the cells with phosphate-buffered saline were washed twice and cultured in Earle’s balanced salt solution (Thermo Fisher, Waltham, MA, USA), with 5.5 mM glucose, without amino acids, for seven hours.

### 4.3. RNA Extraction and miRNA Quantitative Real-Time PCR

RNA was extracted from cells using TRIzol reagent (Invitrogen). miRNA quantitative real-time PCR (qPCR) was performed using a TaqMan miRNA reverse transcription kit, and via an miRNA assay with an ABI PRISM 7000 HT system (Applied Biosystems, CA, USA), following the manufacturer’s instructions. The U48 small nucleolar RNA (RNU48) was chosen as the housekeeping small RNA reference gene. Real-time PCRs were performed in triplicate from three different cDNA samples.

### 4.4. DNA Transfection and RNA Interference

To generate pcDNA3.1-KCNQ1OT1 (36,181–37,140), the fragment of nucleotides from 36,181 to 37,140 of *KCNQ1OT1* was chemically synthesized and ligated into the *BamHI/NotI* site of the pcDNA3.1 vector (Invitrogene). Cells were transfected with pcDNA3.1-HA-CK2α and pcDNA3.1-KCNQ1OT1 (36,181– 37,140) using Polyfect (Qiagen, Hilden, Germany), according to the manufacturer’s instructions. Mimics for miR-760 and control miRNA were purchased from Genolution, Inc. (Seoul, Korea). The antisense inhibitor for miR-760 was obtained from Panagene, Inc. (Seoul, Korea). The siRNA sequence for *CK2**α* was 5′-GAUGACUACCAGCUGGUUCdTdT-3′. The siRNA sequence for *KCNQ1OT1* was 5′–GCCAAUGGAUAGAGAGCAAdTdT-3′. The siRNA sequence for the negative control was 5′-GCUCAGAUCAAUACGGAGAdTdT-3′. RNAs were transfected into cells using Lipofectamine (Invitrogen) for 48 h.

### 4.5. SA-β-gal Activity Assay

Cells in sub-confluent cultures were washed with ice-cold phosphate-buffered saline (PBS), fixed with 3% (*v*/*v*) formaldehyde in PBS for 10 min at room temperature, and then incubated with a staining solution containing 1 mg/mL 5-bromo-4-chloro-3-indolyl β-D-galactoside, 40 mM citric acid-sodium phosphate (pH 6.0), 5 mM potassium ferricyanide, 5 mM potassium ferrocyanide, 150 mM NaCl, and 2 mM MgCl_2_ for 24 h at 37 °C. Blue-stained cells were counted in at least ten fields at 400 × magnification and expressed as a percentage of positive cells.

### 4.6. Measurement of Intracellular ROS Levels

The oxidation-sensitive fluorescent probe CM-H_2_DCFDA was used to evaluate ROS levels. Cells were treated with 5 μM CM-H_2_DCFDA for 20 min at 37 °C in the dark, detached by trypsinization, and then washed with PBS. Fluorescence intensity was determined using a Coulter Elite ESP Cell Sorter (Beckman Coulter Inc., Brea, CA, USA). The forward- and side-scatter gates were set to exclude any dead cells from the analysis; at least 10,000 events within the gate were acquired per sample.

### 4.7. Reverse Transcription (RT)-PCR

Total RNA was extracted from cells. RNA was reverse transcribed using gene-specific primers and reverse transcriptase (Takara Bio Inc., Japan), and the resulting cDNA was PCR amplified. The primers used for the assays are listed in Appendix A. The PCR products were resolved on a 1.5% agarose gel. Quantification of the reverse transcription-PCR bands was performed using densitometry. β-Actin RNA levels were used to standardize the amount of RNA in each sample.

### 4.8. Immunoblotting

Proteins were separated on 10% polyacrylamide gels in the presence of SDS and then transferred onto nitrocellulose membranes. The membranes were blocked with 5% (*w*/*v*) non-fat, dried skim milk in TBST (20 mM Tris-HCl (pH 7.4), 150 mM NaCl, and 0.05% Tween 20) for 2 h and then incubated with specific antibodies in 1% (*w*/*v*) non-fat, dried skim milk for 1 h. The membranes were washed thrice with TBST and then treated with the ECL system to develop the signals (GE Healthcare, Little Chalfont, UK). When deemed necessary, membranes were stripped with stripping buffer (2% SDS, 100 mM β-mercaptoethanol, and 50 mM Tris-HCl (pH 7.0)) at 50 °C for 1 h with gentle shaking and then re-probed with anti-β-actin antibody as an internal loading control.

### 4.9. Statistical Analysis

One-way analysis of variance testing of the data was conducted using the SPSS package program (IBM, Armonk, NY, USA). The results were considered significant if the *p*-value was <0.05. Duncan’s multiple range test was conducted if the differences between the groups were identified as α = 0.05.

## Figures and Tables

**Figure 1 ijms-23-01888-f001:**
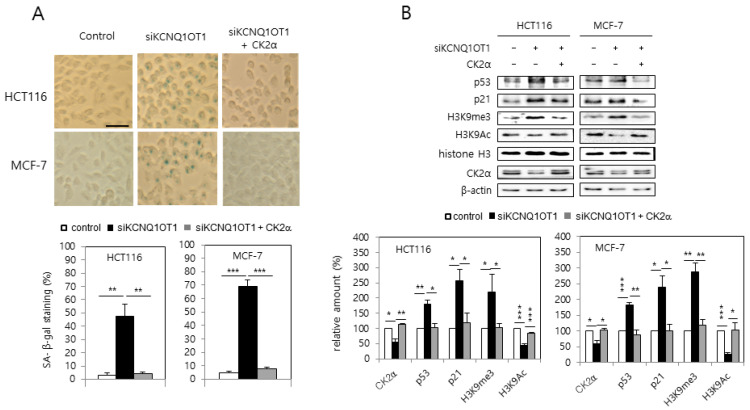
*KCNQ1OT1* knockdown induced activation of SA-β-gal staining, the p53-p21^Cip1/WAF1^ pathway, and H3K9 trimethylation via CK2α silencing in human cancer cells. HCT116 and MCF-7 cells were transfected with *KCNQ1OT1* siRNA for two days in the absence or presence of pcDNA3.1-HA-CK2α. (**A**) Cells were stained with 5-bromo-4-chloro-3-indolyl-β-D-galactoside, and representative images were obtained at 20× magnification (**upper**). Scale bar = 100 μm. Representative data from three independent experiments are shown. The graphs represent the percentage of blue-stained cells (**bottom**). (**B**) Immunoblotting was used to determine the level of each protein using specific antibodies (**upper**). β-Actin was used as a control. The graphs represent the quantitation of each protein relative to β-actin (**bottom**). Data are reported as mean ± SEM. * *p* < 0.05; ** *p* < 0.01; *** *p* < 0.001. H3K9me3, histone H3 Lys9 trimethylation; H3K9Ac, H3 Lys9 acetylation.

**Figure 2 ijms-23-01888-f002:**
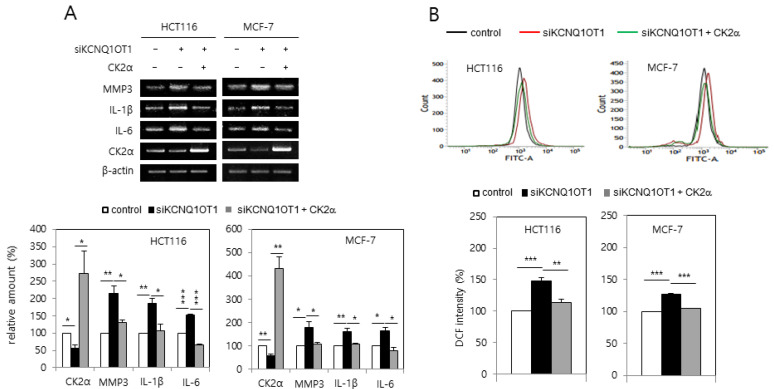
*KCNQ1OT1* knockdown induced senescence-associated secretory phenotype (SASP) factor expression and reactive oxygen species (ROS) generation via *CK2α* silencing in human cancer cells. HCT116 and MCF-7 cells were transfected with *KCNQ1OT1* siRNA for two days in the absence or presence of pcDNA3.1-HA-CK2α. (**A**) The level of each mRNA was determined by reverse transcription-polymerase chain reaction (RT-PCR) using specific primers (**upper**). Representative data from three independent experiments are shown. The graphs represent the quantitation of each mRNA relative to β-actin (**bottom**). (**B**) The cells were incubated with 10 μM CM-H_2_DCFDA. Fluorescence intensity was determined by flow cytometry analysis (**upper**). Representative data from three independent experiments are shown. The graphs show the relative fluorescence level (**bottom**). Data are reported as mean ± SEM. * *p* < 0.05; ** *p* < 0.01; *** *p* < 0.001.

**Figure 3 ijms-23-01888-f003:**
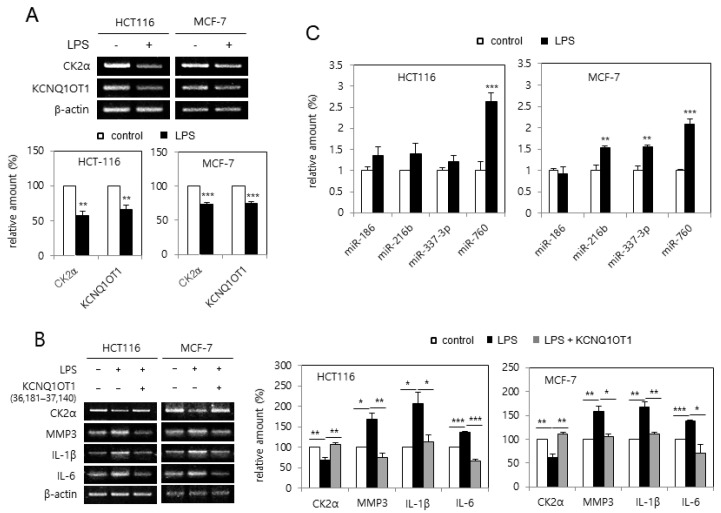
*KCNQ1OT1* was involved in lipopolysaccharide (LPS)-mediated senescence-associated secretory phenotype (SASP) factor expression via silencing *CK2α* in human cancer cells. (**A**) HCT116 and MCF-7 cells were treated with LPS (6 μg/μL) for two days. The level of each mRNA was determined by RT-PCR using specific primers (**top**). Representative data from three independent experiments are shown. β-Actin was used as a control. The graphs represent the quantitation of each mRNA relative to β-actin (**bottom**). (**B**) Cells were treated with LPS (6 μg/μL) in the absence or presence of pcDNA3.1-KCNQ1OT1 (36,181–37,140) for two days. The level of each mRNA was determined by RT-PCR using specific primers (**left**). Representative data from three independent experiments are shown. β-Actin was used as a control. The graphs represent the quantitation of each mRNA relative to β-actin (**right**). (**C**) Cells were treated with LPS (6 μg/μL) for two days. Total RNA was isolated from cells and subjected to qPCR analysis to determine the relative levels of miR-760, miR-186, miR-337-3p, and miR-216b using RNU48 for normalization. Data are shown as the means ± SEM. * *p* < 0.05; ** *p* < 0.01; *** *p* < 0.001.

**Figure 4 ijms-23-01888-f004:**
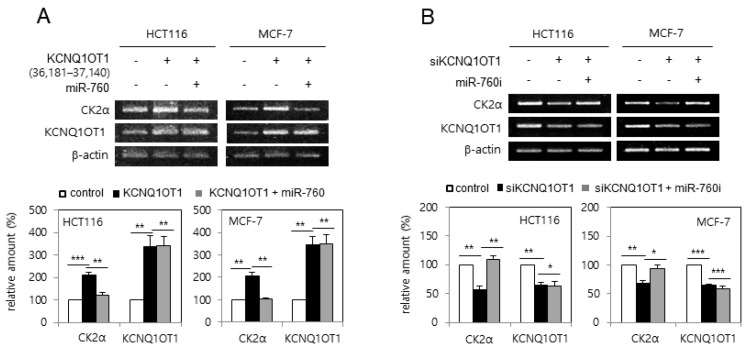
*KCNQ1OT1* upregulated *CK2α* by sponging miR-760 in human cancer cells. (**A**) HCT116 and MCF-7 cells were transfected with pcDNA3.1-KCNQ1OT1 (36,181–37,140) for two days in the absence or presence of miR-760. (**B**) HCT116 and MCF-7 cells were transfected with KCNQ1OT1 siRNA for two days in the absence or presence of miR-760 inhibitor. The level of each mRNA was determined by RT-PCR using specific primers (**top**). Representative data from three independent experiments are shown. The graphs represent the quantitation of each mRNA relative to β-actin (**bottom**). Data are reported as mean ± SEM. * *p* < 0.05; ** *p* < 0.01; *** *p* < 0.001.

**Figure 5 ijms-23-01888-f005:**
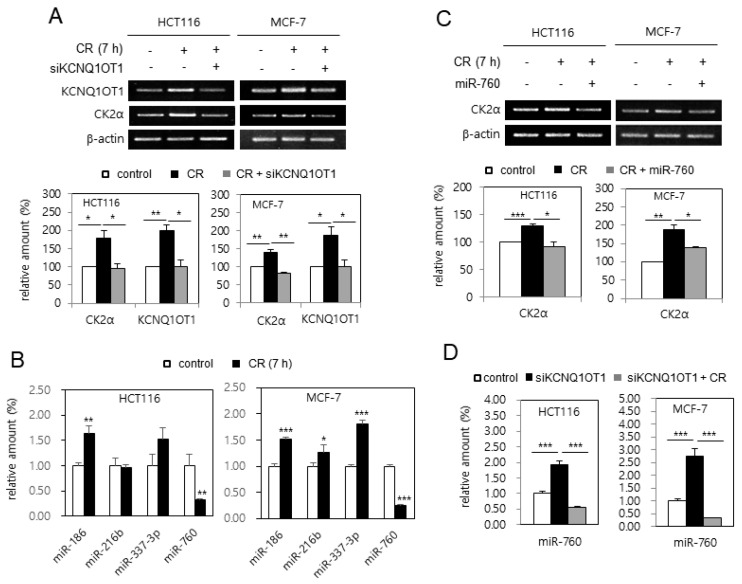
Calorie restriction (CR) condition upregulated CK2α by miR-760 downregulation via *KCNQ1OT1* upregulation in human cancer cells. (**A**,**C** and **D**) HCT116 and MCF-7 cells were transfected with *KCNQ1OT1* siRNA (**A** and **D**) or miR-760 (**C**) for 1.5 days and then incubated under CR conditions for seven hours. (**B**) HCT116 and MCF-7 cells were incubated under CR conditions for seven hours. (**A**,**C**) The level of each mRNA was determined by RT-PCR using specific primers (**top**). Representative data from three independent experiments are shown. β-Actin was used as a control. The graphs represent the quantitation of each mRNA relative to β-actin (**bottom**). (**B**,**D**) Total RNA was isolated from cells and subjected to analysis by qPCR using RNU48 for normalization to determine the relative levels of the indicated miRNAs. Data are shown as the means ± SEM. * *p* < 0.05; ** *p* < 0.01; *** *p* < 0.001.

**Figure 6 ijms-23-01888-f006:**
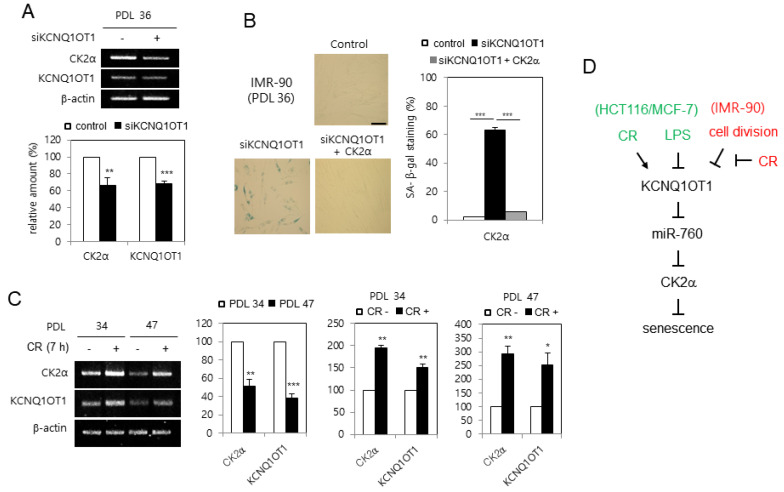
*KCNQ1OT1* was downregulated during replicative senescence in human lung fibroblast cells, which was able to be rescued by calorie restriction (CR) conditions. (**A**) IMR-90 cells (PDL 36) were transfected with *KCNQ1OT1* siRNA. The level of each mRNA was determined by RT-PCR using specific primers (**top**). Representative data from three independent experiments are shown. β-Actin was used as a control. The graphs represent the quantitation of each mRNA relative to β-actin (**bottom**). (**B**) IMR-90 cells (PDL 36) were transfected with *KCNQ1OT1* siRNA for two days in the absence or presence of pcDNA3.1-HA-CK2α. Cells were stained with 5-bromo-4-chloro-3-indolyl-β-D-galactoside, and representative images were obtained at 20× magnification (**left**). Scale bar = 100 μm. Representative data from three independent experiments are shown. The graphs represent the percentage of blue-stained cells (**right**). (**C**) IMR-90 cells of PDL 34 and PDL 47 were incubated under CR conditions for seven hours. The level of each mRNA was determined by RT-PCR using specific primers (**left**). Representative data from three independent experiments are shown. β-Actin was used as a control. The graphs represent the quantitation of each mRNA relative to β-actin (**right**). Data are reported as mean ± SEM. * *p* < 0.05; ** *p* < 0.01; *** *p* < 0.001. (**D**) Possible model illustrating the roles of *KCNQ1OT1* for senescence and CR. PDL, population doubling level.

## Data Availability

Not applicable.

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
