# Peer review of "Long Non-Coding RNA KCNQ1OT1 Regulates Protein Kinase CK2 Via miR-760 in Senescence and Calorie Restriction"

_ijms, 2022, doi:10.3390/ijms23031888_

Round 1
Reviewer 1 Report
The authors must provide full western blots, not cropped images. I would be happy to review the manuscript once the authors provide full, uncropped western blot images.

Author Response
In this study, we transferred the electrophoresed proteins to a nitrocellulose film, cut the films based on the expected molecular weights, and then performed western blotting using specific antibodies against each protein. This procedure was performed to keep the loading amounts of proteins constant for each sample. If different blots had been used for the western blotting of each protein, the protein amounts in samples would not be constant. Particularly, the protein amount can be accurately determined by comparing it with the β-actin protein loaded on the same blot. Furthermore, because every experiment has been repeated at least three times in this study, we are confident of the results presented in this paper.

Reviewer 2 Report
In this manuscript, Lee and Bae studied role of KCNQ1OT1 in senescence phenotype via downregulating CK2α in human cancer MCF-7 and HCT116 cells and lung fibroblast IMR-90 cells
Results concluded that KCNQ1OT1 upregulated CK2α expression through interaction with miR760 in senescence and calorie restriction which suggest this long non coding RNA may be a unique therapeutic target for aging associated diseases. Cell cycle arrest is often used bycells to facilitate DNA repair before cell proliferation. If, however, the DNA damage is too severe, other signaling mechanisms work to induce cell senescence or apoptosis to prevent malignancies. Did you tested cellcycle analysis or apoptosis or dna damage markers.
a. KCNQ1OT1 was involved in lipopolysaccharide mediated SASP factor expression via silencing CK2α in cancer cells. b. KCNQ1OT1 was downregulated during replicative senescence in human lung fibroblast cells, which was rescued by calorie restriction condition. Here a and b used different type of cell lines. explain the reason.
what is your justification to choose these specific cell type in this study.
fig1a and 6b scale bar is missing.
materials: antibody cat.# need to be added
can be focus more detail in epigenetic machinery
all the experiments were performed by only one researcher. I didnt see any acknowledgement to any research assistance. It looks similar to me some of the blots . for eg. in these IL1 beta: Fig.2a MCF7 and fig3b HCT116. verify this.
Overall well written manuscript with detailed description.
Author Response
In this manuscript, Lee and Bae studied role of KCNQ1OT1 in senescence phenotype via downregulating CK2α in human cancer MCF-7 and HCT116 cells and lung fibroblast IMR-90 cells.
Results concluded that KCNQ1OT1 upregulated CK2α expression through interaction with miR760 in senescence and calorie restriction which suggest this long non coding RNA may be a unique therapeutic target for aging associated diseases. Cell cycle arrest is often used by cells to facilitate DNA repair before cell proliferation. If, however, the DNA damage is too severe, other signaling mechanisms work to induce cell senescence or apoptosis to prevent malignancies. Did you tested cell cycle analysis or apoptosis or dna damage markers.
We previously reported that CK2 inhibition triggers the expression of several senescence markers, including senescence-associated β-galactosidase (SA-β-gal) activity, p53–p21Cip1/WAF1 axis activation, reactive oxygen species (ROS) production, senescence-associated heterochromatin foci (SAHF) formation, and senescence-associated secretory phenotype (SASP) expression. Therefore, we focused on the role of lncRNA KCNQ1OT1 for SA-β-gal staining, ROS production, SAHF formation, and SASP expression in the present study. According to your suggestion, we would like to investigate cell cycle analysis, apoptosis, and DNA damage markers in future studies.
KCNQ1OT1 was involved in lipopolysaccharide mediated SASP factor expression via silencing CK2α in cancer cells. b. KCNQ1OT1 was downregulated during replicative senescence in human lung fibroblast cells, which was rescued by calorie restriction condition. Here a and b used different type of cell lines. explain the reason.
As senescence suppresses tumor progression, it is believed to be an efficient anticancer therapy. Thus, human cancer HCT116 and MCF-7 cell lines were used in this study. In addition, IMR-90 cells were used to investigate whether the phenomenon observed in cancer cells also occurs in the replicative senescence of normal fibroblasts. Data in Figure 3 indicate that KCNQ1OT1 is involved in lipopolysaccharide-mediated CK2α downregulation and SASP expression in cancer cells. Data in Figure 5 indicate that calorie restriction upregulates KCNQ1OT1, resulting in CK2α upregulation in cancer cells. Finally, Figure 6 indicates that the phenomenon observed in cancer cells also occurs in the replicative senescence of normal fibroblasts.
what is your justification to choose these specific cell type in this study.
As senescence suppresses tumor progression, it could be an efficient anticancer therapy. Therefore, human cancer HCT116 and MCF-7 cell lines were used in this study. In addition, IMR-90 cells were used to investigate whether the phenomenon observed in cancer cells also occurs in the replicative senescence of normal fibroblasts.
fig1a and 6b scale bar is missing.
According to your suggestion, we have added scale bars in Figs. 1A and 6B.
materials: antibody cat.# need to be added
We have added the catalog numbers for all antibodies according to your suggestion. (page 9)
can be focus more detail in epigenetic machinery
We have added the following sentence in the “Discussion” section with the appropriate reference citation according to your suggestion. (pages 7 and 11)
Epigenetic regulations, including microRNAs (miRNAs), nucleosome remodeling, DNA methylation, and histone modification, can change heritable phenotypes without changing DNA sequences [26].
26. Ramzan, F.; Vickers, M.H.; Mithen, R.F. Epigenetics, microRNA and metabolic syndrome: a comprehensive review. Int. J. Mol. Sci. 2021, 22, 5047.
all the experiments were performed by only one researcher. I didnt see any acknowledgement to any research assistance. It looks similar to me some of the blots . for eg. in these IL1 beta: Fig.2a MCF7 and fig3b HCT116. verify this.
The first author conducted all experiments in this study.
We confirmed that all blots in Fig. 2A are different from those in Fig. 3B.
Overall well written manuscript with detailed description.
Thank you for your comments.

Reviewer 3 Report
manuscript is concise and clerar
Well presented and documented
Now is ready to be published
Author Response
manuscript is concise and clerar
Well presented and documented
Now is ready to be published
Thank you for your comments.

Round 2
Reviewer 1 Report
Please include all the original images for gels/blots as supplemental figures.
Author Response
Thank you for your comments.